# The Balance between the Left and Right Ventricular Deformation Evaluated by Speckle Tracking Echocardiography Is a Great Predictor of the Major Adverse Cardiac Event in Patients with Pulmonary Hypertension

**DOI:** 10.3390/diagnostics12092266

**Published:** 2022-09-19

**Authors:** Xueming Zhang, Binqian Ruan, Zhiqing Qiao, Menghui Yang, Qi Zhuang, Jian Wang, Wei Wang, Ying Zheng, Hang Zhao, Xuedong Shen, Jieyan Shen

**Affiliations:** 1Department of Cardiology, Renji Hospital, School of Medicine, Shanghai Jiao Tong University, Shanghai 200127, China; 2Department of Internal Medicine, Shanghai Changzheng Hospital, Second Military Medical University, Shanghai 200003, China

**Keywords:** biventricular balance, speckle tracking, pulmonary hypertension, prognosis

## Abstract

Cardiovascular failure is one of the most relevant causes of death in pulmonary hypertension (PH). With progressive increases of right ventricular (RV) afterload in PH patients, both RV and left ventricular (LV) function impair and RV–LV dyssynchrony develop in parallel. We aimed to analyze the balance between the left and right ventricular deformation to assess the outcome of patients with pulmonary hypertension by means of speckle tracking echocardiography. In this prospective study, 54 patients with invasively diagnosed pulmonary hypertension, and 26 healthy volunteers were included and underwent a broad panel of noninvasive assessment including 2D-echocardiography, 2D speckle tracking, 6-minute walking test and BNP. Patients were followed up for 338.7 ± 131.1 (range 60 to 572) days. There were significant differences in |LVGLS/RVFLS-1| and |LASc/RASc-1| between PH patients and the control group. During the follow up, 13 patients experienced MACEs, which included 7 patients with cardiac death and 6 patients with re-admitted hospital due to right ventricular dysfunction. In the multivariate Cox model analysis, |LVGLS/RVFLS-1| remained independent prognosis of markers (HR = 4.03). Our study findings show that |LVGLS/RVFLS-1| is of high clinical and prognostic relevance in pulmonary hypertension patients and reveal the importance of the balance between the left and right ventricular deformation.

## 1. Introduction

Pulmonary hypertension (PH) is a group of diseases characterized by persistent progressive pulmonary arteriolar wall thickening, luminal stenosis and elevated pulmonary vascular resistance. Continuously excessive afterload of the right ventricle and right heart failure induced by the increased pulmonary artery resistance is the main cause of death in most patients with end-stage pulmonary hypertension [1,2]. Right ventricular (RV) dysfunction in patients with PH has a vital influence on longevity [3] and protection of RV function is crucial in management and follow-up of PH patients [4,5].

Currently, echocardiography is the most widely available technique for evaluating the performance of the RV. Patients with PH can benefit from the speckle tracking echocardiography for risk stratification by evaluating RV systolic function [6,7,8]. However, increasing right ventricular pressure overload and dilatation of RV not only affects left ventricular (LV) diastolic function [5,9,10], but also leads to LV contractile dysfunction [11,12,13,14], because of the common interventricular septum, pericardium and myocardial fibrosis. We hypothesized that the left and right ventricles’ deformation is interactive, and balance of the deformation between the left and right ventricles is related to the outcome in patients with pulmonary hypertension.

## 2. Materials and Methods

### 2.1. Study Population and Design

The study population consisted of 54 patients with pulmonary hypertension diagnosed by right heart catheterization in the East and South Renji Hospitals affiliated to Shanghai Jiao Tong University School of Medicine from September 2018 to July 2020. The enrollment criteria for the pulmonary hypertension group included: aged 18–80 years old, with a resting mean pulmonary artery pressure (mPAP) ≥ 25 mmHg. Exclusion criteria included pulmonary hypertension due to left heart disease, hypoxic pulmonary hypertension, patients with severe complications (such as tumors) and atrial fibrillation, pregnancy, poor quality echocardiographic images and inability to complete follow-up. Another 26 healthy volunteers who were age and gender matched were recruited as the control group. All PH patients also underwent a 6-minute walking test and B-type natriuretic peptide (BNP) measurement.

### 2.2. Echocardiography

Echocardiography examinations in PH patients were performed on the same day of the 6-minute walking test and BNP measurement. Two-dimensional and four-dimensional (4D) echocardiographic images were acquired by an experienced physician using Vivid E9 scanner (GE Health, USA), equipped with 4V and M5S ultrasound probes (1.5 to 4.5 MHz) and synchronously connected ECG. The EchoPAC 201.0 version software (GE Health, USA) was used for offline analysis for the stored images by two physicians who were blinded to the patients’ condition. All parameters were the mean measurements of three consecutive cardiac cycles during a single breath-hold.

Echo recordings and measurements followed the current guidelines [15,16]. Measurements of two-dimensional echocardiography included tricuspid annular plane systolic excursion (TAPSE), RV fractional area change (RVFAC), right ventricular systolic pressure (RVSP) estimated through tricuspid regurgitation and left ventricular ejection fraction (LVEF) through biplane Simpson’s method both in the apical four-chamber view and apical two-chamber view.

Two-dimensional speckle tracking echocardiographic analysis was performed from apical four-chamber view with the clear ventricular or atrial wall. Measurements of two-dimensional speckle tracking echocardiography included left ventricular global longitudinal strain (LVGLS), right ventricular free-wall longitudinal strain (RVFLS), left atrial reservoir function (LASr), left atrial conduit function (LASc), left atrial pump function (LASp), right atrial reservoir function (RASr), right atrial conduit function (RASc) and right atrial pump function (RASp). The endocardial boundary is delineated manually, and the width of spot tracking is adjusted appropriately to match the thickness of the ventricular wall or atrial wall. Images with more than two tracking-failed segments will be discarded.

### 2.3. 6MWD and Borg Dyspnea Scale

The 6-minute walking test was conducted according to American Thoracic Society guidelines [17]. The test was carried out on a 30-meter-long, flat, straight, enclosed corridor. SpO2, blood pressure and heart rate before and after the test were measured and recorded. The Borg dyspnea scale was used to assess the degree of dyspnea in patients after the 6-minute walking test.

### 2.4. Follow-Up Assessment

All patients were followed up by phone or outpatient clinic every three months to record the occurrence of the endpoint events. A major adverse cardiac event (MACE) was defined as all-cause death and readmission due to right heart failure, pulmonary arterial hypertension complications or clinical deterioration.

### 2.5. Statistical Analysis

Continuous variables were expressed as the mean ± SD for normally distributed data and as median with interquartile range when variables were skew distributed. An independent samples t-test was used for comparison between two groups with normal distribution. The Mann–Whitney U test was used to compare the data with the skewed distribution. Correlation between two variables was assessed by Pearson correlation coefficient. Variables that achieved a significant level of *p* < 0.05 by Cox proportional univariate hazard model were re-evaluated using a Cox proportional multivariate hazard model for determine of the independent predictor of the endpoint events in PH patients.

Moreover, we introduced the new parameters |LVGLS/RVFLS-1|, |LASr/RASr-1|, |LASc/RASc-1| and |LASp/RASp-1|, which were the absolute values of the ratio of left and right ventricular or atrial motion parameters minus 1 to represent the deviation of LVGLS/RVFLS, LASr/RASr, LASc/RASc and LASp/RASp from 1 to evaluate the motion balance between left and right heart.

A *p* value of <0.05 was considered statistically significant. All statistical analysis was performed using IBM SPSS 19.0.

## 3. Results

The follow up duration in all PH patients was 338.7 ± 131.1 (range 60 to 572) days. During the follow-up period, 13 patients had MACE, which included 7 patients with cardiac death and 6 patients with re-admitted hospital due to right ventricular dysfunction.

### 3.1. Clinical Characteristics

The demographic, etiology, clinical characteristics and invasive hemodynamics of the patients are noted in Table 1. The majority of patients were female (87.04%) and the average age was 44.58 ± 15.45 years. This cohort consisted of 16 (29.63%) patients with idiopathic pulmonary arterial hypertension and 24 (44.44%) pulmonary arterial hypertension patients associated with connective tissue disease. In Table 1, it is shown that there is no significant difference between the 27 healthy volunteers and PH patients in gender, age and BSA (*p* = 0.60–0.97).

Echocardiography-derived conventional parameters included TAPSE (24.50 (22.00, 26.00) vs. 16.00 (13.00, 18.00), *p* = 0.000) and FAC (43.62 ± 6.73 vs. 24.77 ± 8.46, *p* = 0.000), which were significantly decreased in PH patients compared to controls, but not in LVEF (61.11 ± 3.62 vs. 59.50 ± 7.47, *p* = 0.195) (Table 1). LVGLS and RVFLS in the control group were −18.05 ± 2.76% and −19.90 ± 5.43%, respectively. LVGLS was a little bit less than RVFLS, but there was no significant difference between LVGLS and RVFLS (*p* > 0.05). Compared with the control group, LVGLS (−18.28 (−20.00, −15.38) vs. −15.70 (−17.68, −12.80), *p* = 0.004) and RVFLS (−19.90 ± 5.43 vs. −13.02 ± 5.02, *p* < 0.001) were significantly decreased in PH patients (The minus sign“−” represents the direction of ventricular longitudinal contraction vector and has no mathematical significance). Both left and right atrial reservoir and conduction were markedly reduced with the numerically largest difference in conduct function. Although left atrial pump function in PH patients was slightly lower than controls, there was no significant difference in right atrial pump function between PH patients and the control group.

### 3.2. Prediction of MACE-Free Survival by LVGLS and RVFLS

Figure 1 depicts Kaplan–Meier survival analysis for MACEs among four groups of PAH patients stratified by the median value of LVGLS and RVFLS. MACE-free survival in patients with combination of LVGLS > −15.59 and RVFLS > −13.75 (both preserved group) was significant better than those with LVGLS ≤ −15.59 and RVFLS ≤ −13.75 (both impaired group) (*p* < 0.05). Although the difference between patients with both preserved or both impaired and patients with LVGLS impaired or RVFLS impaired was not significant because of the small sample size of the LVGLS impaired group and RVFLS impaired group, it can be seen from the K-M survival curve of each group that survival rate of patients reduced whether LVGLS alone or RVFLS alone decreased. Additionally, when both decrease, the survival rate of patients will decrease more significantly.

Based on the trend shown in Figure 1, we speculated that only LVGLS or RVFLS might not be enough to predict the prognosis of patients, and the relative balance of LVGLS and RVFLS may be more important than only LVGLS or RVFLS preserved. In normal people, their LVGLS/RVFLS approached 1. However, in PH patients, not only did RVFLS decrease significantly, but LVGLS also decreased in some PAH patients, and the decrease was more obvious than RVFLS, so that LVGLS/RVFLS of PH patients may be higher or lower than 1. Thus, we introduced a new index reflecting balance between left and right ventricular longitudinal deformations, |LVGLS/RVFLS-1|, which is the absolute value of the ratio of left and right ventricular longitudinal strain minus 1, considering the balance between left and right atrial reservoir (|LASr/RASr-1|), conduit (|LASc/RASc-1|) and pump (|LASp/RASp-1|) function.

### 3.3. Balance between Left and Right Atrial and Ventricular Longitudinal Deformations in PH Patients

As Table 2 showed that |LVGLS/RVFLS-1| in the control group was 0.20 ± 0.17. |LASr/RASr-1|, |LASc/RASc-1| and |LASp/RASp-1| in the control group were 0.23 (0.09, 0.67), 0.34 ± 0.18 and 0.33 (0.20, 0.39), respectively. Like ventricular and atrial longitudinal strain, there were significant differences in |LVGLS/RVFLS-1| and |LASc/RASc-1| between PH patients and the control group.

### 3.4. Correlation between |LVGLS/RVFLS-1| and Clinical Measurements or Echocardiography-Derived Parameters

As Table 3 showed that |LVGLS/RVFLS-1| correlated with RVFLS (r = 0.559, *p* < 0.0001) and BNP (r = 0.4125, *p* = 0.0036), RASr (r = −0.3766, *p* = 0.0054), RASc (r = −0.3579, *p* = 0.0085), RVSP (r = 0.3385, *p* = 0.0132) and FAC (r = −0.3082, *p* = 0.0248). However, there was no significant relation between |LVGLS/RVFLS-1| and 6MWD.

### 3.5. Prediction of Ominous Prognosis

Univariate analysis demonstrated that BNP, 6MWD, NYHA, FAC, RVSP, LVGLS, RVFLS, LASc, RASr, |LVGLS/RVFLS-1| and |LASp/RASp-1| were the most significant predictors of the MACE (Table 4). The Hazard ratio of |LVGLS/RVFLS-1| was the highest (HR = 4.03) and associated with a 4.03-fold increase in MACE. After adjustment for age, gender and the variables with significant difference by Cox univariate regression, the |LVGLS/RVFLS-1| increase and 6MWD decrease were continuously related to MACE in the multivariate Cox analysis. The |LVGLS/RVFLS-1| increase was associated with a 3.15-fold increase in MACE. The Hazard ratio of |LVGLS/RVFLS-1| (HR = 3.15) was significantly higher than 6MWD (HR = 0.995).

## 4. Discussion

Kado et al. reported that the four-chamber longitudinal strains in patients with cardiac amyloidosis were significantly associated with increase of major adverse cardiovascular events, with an incremental value over traditional echocardiographic parameters, and the relative preservation of both LV LS and RA LS values may better than the reduction in one or more strain values on prognosis of the patients [18]. According to the theory of Chinese Traditional Medicine, everybody situates as a state of YIN (negative) and YANG (positive) balance [19], and they will be ill if this balance is lost [20]. We hypothesized that there should be a dynamic balance situated in between the functions of the four cardiac chambers. Therefore, we conducted this study, and the results validated the importance of the balance between the left and right ventricular deformation.

### 4.1. Significance of Baseline Parameters

In our study, compared to the control group, the parameters reflecting right and left ventricular and atrial function of pulmonary hypertension patients were decreased except right atrial systolic function and LVEF, and both right and left atrial conduit function were severely decreased. Decreased RA conduit function may result from RV delayed relaxation and diastolic dysfunction because of the RV afterload increment and hypertrophy of the RV myocardium. Reduced reservoir function and increased systolic function can be explained by the Frank-Starling law [21]: With the increase of preload, the initial length of atrium increases, so that the atrial pump function is improved. The relative enhancement of active contraction of right atrium may also be a compensatory mechanism of right heart failure to compensate for the insufficiency of right ventricular diastolic filling. At the same time, the increase of right ventricular end diastolic pressure also reduces the amount of blood returned to the atrium, resulting in decrease of the reserve function of the right atrium. This result is consistent with some previous research results [22,23]. Although PAH influences RA function directly, LA reservoir, conduit and systolic function compared with control group were also impaired. LA dysfunction may be due to right atrial dilatation and compression.

We always consider the right heart and left heart as two independent chambers, but this distinction is somewhat artificial. Not only does the circumferential myocardium on the superficial right ventricular wall extend to the left ventricle, but also both ventricles share ventricular septum and pericardium, completely synchronized electrophysiological properties. As we all know, electrical dyssynchrony which leads to remodeling of ventricles with a faster increase of radial and axial fiber growth [24]. It will be interesting to explore if there is an interaction effect between electrophysiological remodeling and dyssynchrony mechanics in PH patients. In addition, several derangements occur in the RV muscle that ultimately contribute to neuroendocrine microenvironment impairment, which may be primarily mediated by oxidative stress and pro-inflammatory signaling [25]. Impaired RVFLS of the PH patients was obvious; however, a moderate decrease in the LVFLS was also observed. Moreover, linear regression analysis found that the correlation between RVFLS and LVGLS. In our study, echocardiography revealed that geometry of the left ventricle was altered because of right ventricular enlargement. Some experimental animal and clinical studies demonstrated that one of complications of severe pulmonary hypertension is atrophic remodeling of the left ventricle [11,26,27,28,29]. Other studies have shown that compared with the control group, not only LV global longitudinal strain, but also radial strain [30] and septal circumferential strain decreased [31,32]. Two previous studies have also reported that PAH patients had reduced LV longitudinal strain despite normal LVEF [33,34]. Our research study yielded similar results that left ventricular longitudinal strain decreased even if left ventricular ejection fraction was normal in PAH patients (LVEF (61.11 ± 3.62 vs. 59.50 ± 7.47, *p* = 0.195), LVGLS (−18.28 (−20.00, −15.38) vs. −15.70 (−17.68, −12.80), *p* = 0.004)). These studies demonstrated that there is a close relationship between the left and right heart, and echocardiographic speckle tracking is more sensitive than LVEF in early detection of left ventricular impairment.

### 4.2. Biventricular Balance

At first, we conducted a Kaplan–Meier survival analysis for MACEs among four groups of PAH patients stratified by the median value of LVGLS and RVFLS. The curve showed that not only the survival rate of patients with impaired right ventricular function will decrease, but also the survival rate of patients with impaired left ventricular function, and MACE-free survival was similarly worse in patients if either LVGLS or RVFLS was below the cutoff values, which is similar to Kado’s study [18]. Therefore, we postulated that cardiac chamber function balance may play an important role in the prognosis of patients and introduce new parameters to further explore.

Thus, we introduced a few new parameters including |LVGLS/RVGLS-1|, |LASr/RASr-1|, |LASc/RASc-1| and |LASp/RASp-1| to represent the balance between left and right ventricular global longitudinal strain and left and right atrial functions. In PAH patients, similar to RVGLS, LVGLS and RASc, LASc, |LVGLS/RVFLS-1| and

|LASc/RASc-1| were significantly increased, representing a greater deviation between left and right ventricular global longitudinal strain and atrial conduit function.

Furthermore, the results showed significant correlations between |LVGLS/RVGLS-1| and BNP, NYHA, FAC and RVSP, but |LVGLS/RVGLS-1| did not have relationship with 6MWD, which has shown a prognostic effect in previous studies. 6MWD is an indicator that can reflect the activity tolerance of patients, while |LVGLS/RVGLS-1| represents the biventricular movement balance. We speculated the reason of this result is that the left and right ventricular motion began to be unbalanced before the 6MWD was decreased, and the changes of the two were not synchronous.

Our small sample study showed that |LVGLS/RVFLS-1| and |LASp/RASp-1| were an important ratio in outcome of patients with pulmonary hypertension, especially |LVGLS/RVFLS-1|. |LVGLS/RVFLS-1| was associated with a 4.03-fold increase in MACE in Cox univariate regression. After adjustment for age and the variables with significant difference by Cox univariate regression, |LVGLS/RVFLS-1| increases were continuously related to MACE in the multivariate Cox analysis. The |LVGLS/RVFLS-1| increase was associated with a 3.15-fold increase in MACE. The Hazard ratio of |LVGLS/RVFLS-1| (HR = 3.15) was significantly higher than 6MWD (HR= 0.995). The results showed that the balance between the left and right ventricular longitudinal strain was important in patients with pulmonary hypertension. If the patient loses the balance, the prognosis of the patient will be worse. Our study was the first presented the impact of |LVGLS/RVFLS-1| in patients with pulmonary hypertension, which provided us a completely new, simple and great predictor for the evaluation of patient’s outcome. We hope that this parameter could be used not only in pulmonary hypertension patients, but also in any patients with the disease caused unbalanced left and right ventricular movements, such as electrophysiological remodeling mechanisms.

Univariate and multivariate COX regression analysis showed that only 6MWD and |LVGLS/RVFLS-1| were independent prognostic factors for death or re-admission. Although RV longitudinal peak systolic strain has been to be a significant determinant of all-cause mortality [35,36], another study has shown that RV global strain was not independent predictor of survival [37]. In our multivariate analysis, single indicators, such as LVGLS or RVFLS, did not play a significant role in predicting the prognosis of patients. This may be due to the influence of |LVGLS/RVFLS-1| in multivariate Cox regression, insufficient sample size or insufficient follow-up time, but it also shows that |LVGLS/RVFLS-1| is more sensitive than LVGLS or RVFLS in predicting prognosis of patients.

In a few patients, we observed that the decrease of LVGLS is more obvious than RVFLS because of the compression of right heart, which is different from PAH patients with RVFLS decreased as the main manifestation. These patients were often neglected in previous clinical studies focusing on the right heart function of PAH patients, but their survival rate is also affected by pulmonary hypertension. The new composite index we introduced can include not only the patients with decreased right ventricular function, but also these patients, which may be one of the reasons why the index is more sensitive in the multifactor prognosis analysis.

The results of our study remind us that the interaction and regulation of left and right ventricles in the structure, function, electrophysiological mechanisms and neuroendocrine regulatory system may be more than we expected. We should pay attention to protect left heart function in PAH patients, and preservation and balance of left and right ventricular function may benefit patients more.

### 4.3. Limitations

There are several limitations of this study that need to be pointed out. First, we could not perform subgroup analysis according to PASP, NYHA and so on because of the small sample population. Second, the follow-up time was not long enough, and the number of patients with positive results was small. Third, atrial strain measured by speckle tracking has some deviations due to thin atrial wall. Meanwhile, due to the enlargement of the right heart in patients with pulmonary hypertension, it is difficult to include two ventricles in one section, which may lead to deviation of ejection fraction measured by four-dimensional ultrasound. Lastly, myocardial strain includes longitudinal strain and circumferential strain, but only longitudinal strain was measured in this study, and two-dimensional speckle tracking can only reflect myocardial movement in the same plane, so our results need to be confirmed by three-dimensional speckle tracking.

## 5. Conclusions

Previous studies on the prognosis of pulmonary hypertension patients have focused on right ventricular systolic function or left ventricular diastolic function. Our study revealed that the motion balance between left and right ventricular may be one of the important indexes to evaluate the prognosis of PH patients. The interaction and regulation of left and right ventricles in the structure, function, electrophysiological mechanisms and neuroendocrine regulatory system may be more than we expected.

## Figures and Tables

**Figure 1 diagnostics-12-02266-f001:**
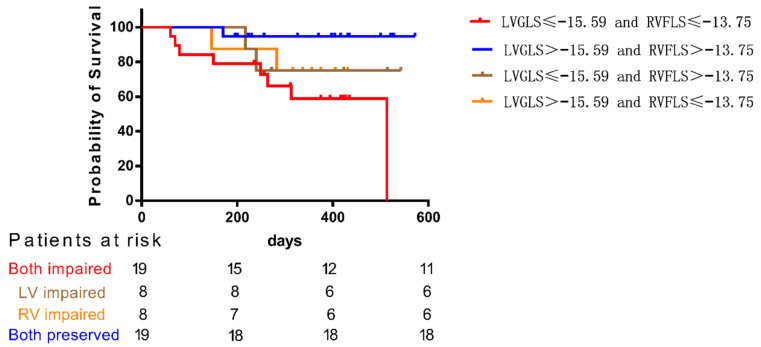
Kaplan–Meier survival analysis for MACEs among four groups of PAH patients stratified by the median value of LVGLS and RVFLS.

**Table 1 diagnostics-12-02266-t001:** Comparison of clinical data between the control group and pulmonary hypertension group.

Variable	Ctrl (N = 26)	PH (N = 54)	*p*
Female (*n* (%))	22 (84.6)	47 (87.04)	0.819
Age (y, x ± s)	44.70 ± 14.65	44.58 ± 15.45	0.974
BSA (m^2^, x ± s)	1.56 ± 0.13	1.54 ± 0.12	0.603
Etiology (%)			
Idiopathic pulmonary arterial hypertension	-	16 (29.63)	
Connective tissue disease	-	24 (44.44)	
Congenital heart disease	-	7 (12.96)	
Chronic thromboembolic pulmonary hypertension	-	6 (11.11)	
Portal hypertension	-	1 (1.85)	
Right heart catheterization			
mPAP (mmHg, x ± s)	-	53.80 ± 13.45	
PCWP[mmHg, M (Q1, Q3)]	-	13.00 (9.00, 15.00)	
PVR (Wood units, x ± s)	-	10.66 ± 4.99	
CI[L/min·m^2^, M (Q1, Q3)]	-	2.50 (2.08, 3.53)	
NYHA classification (%)			
I	-	22 (40.74)	
II	-	16 (29.63)	
III	-	10 (18.52)	
IV	-	6 (11.11)	
BNP[M (Q1, Q3)	-	242.00 (84.50, 622.00)	
6MWD[m, M (Q1, Q3)	-	410.00 (360.00, 459.00)	
Echocardiography			
TAPSE[mm, M (Q1, Q3)]	24.50 (22.00, 26.00)	16.00 (13.00, 18.00)	<0.001
FAC (x ± s)	43.62 ± 6.73	24.77 ± 8.46	<0.001
LVEF (%, x ± s)	61.11 ± 3.62	59.50 ± 7.47	0.195
RVSP (mmHg, x ± s)	-	75.88 ± 25.40	
Speckle tracking			
LVGLS[%, M (Q1, Q3)	−18.28 (−20.00, −15.38)	−15.70 (−17.68, −12.80)	0.004
LVGLS[%, x ± s]	−18.05 ± 2.76	−15.68 ± 4.50	0.004
RVFLS (%, x ± s)	−19.90 ± 5.43	−13.02 ± 5.02	<0.001
LASr (%, x ± s)	36.88 ± 13.45	26.07 ± 10.00	<0.001
LASc[%, M (Q1, Q3)	22.32 (16.88, 29.75)	11.69 (8.22, 18.84)	<0.001
LASp (%, x ± s)	15.00 ± 5.79	12.33 ± 4.62	0.034
RASr (%, x ± s)	31.82 ± 12.87	25.63 ± 13.50	0.014
RASc[%, M (Q1, Q3)	17.18 (10.88, 23.43)	8.02 (5.12, 14.80)	<0.001
RASp (%, x ± s)	15.15 ± 5.92	15.39 ± 8.37	0.899

Data are expressed as the mean value ± SD or median (Q1, Q3) or number (percentage) of patients. mPAP, mean pulmonary artery pressure; PCWP, pulmonary capillary wedge pressure; PVR, pulmonary vascular resistance; CO, cardiac output; 6MWD, 6-minute walking distance; TAPSE, tricuspid annular plane systolic excursion; FAC, fraction area change; LVEF, left ventricular ejection fraction; RVSP, right ventricular systolic pressure; LVGLS, left ventricular global longitudinal strain; RVFLS, right ventricular free-wall longitudinal strain; LASr, left atrial reservoir function; LASc, left atrial conduit function; LASp, left atrial pump function; RASr, right atrial reservoir function; RASc, right atrial conduit function; RASp, right atrial pump function.

**Table 2 diagnostics-12-02266-t002:** Balance between left and right longitudinal deformations.

Variable	Ctrl (N = 26)	PH (N = 54)	*p*
|LVGLS/RVFLS-1|	0.20 ± 0.17	0.27 (0.13, 0.66)	0.009
|LASr/RASr-1|	0.23 (0.09, 0.67)	0.39 (0.19, 0.64)	0.058
|LASc/RASc-1|	0.34 ± 0.18	0.61 (0.19, 1.47)	0.033
|LASp/RASp-1|	0.33 (0.20, 0.39)	0.42 (0.22, 0.62)	0.136

LVGLS, left ventricular global longitudinal strain; RVFLS, right ventricular free-wall longitudinal strain; LASr, left atrial reservoir function; LASc, left atrial conduit function; LASp, left atrial pump function; RASr, right atrial reservoir function; RASc, right atrial conduit function; RASp, right atrial pump function.

**Table 3 diagnostics-12-02266-t003:** Correlation between |LVGLS/RVFLS-1| and clinical measurements or echocardiography-derived parameter.

Variable	Spreaman	Correlation (95% Confidence Interval)	*p*
|LVGLS/RVFLS-1| versus			
BNP	0.4125	0.137–0.6288	0.0036
6MWD	−0.1421	−0.4065–0.1442	0.3148
NYHA	0.2366	−0.04413–0.4828	0.088
FAC	−0.3082	−0.5398–−0.03314	0.0248
TAPSE	−0.2307	−0.4802–0.05326	0.0998
RVSP	0.3385	0.06692–0.5634	0.0132
LVEF	0.03285	−0.2473–0.3079	0.8154
LVGLS	0.02415	−0.2554–0.3	0.8637
RVFLS	0.559	0.3328–0.7243	<0.0001
LASr	−0.1518	−0.4123–0.1316	0.2779
LASc	−0.1656	−0.424–0.1177	0.236
LASp	0.02411	−0.2555–0.3	0.8639
RASr	−0.3766	−0.5925–0.1103	0.0054
RASc	−0.3579	−0.5783–−0.08888	0.0085
RASp	−0.2129	−0.4634–0.06904	0.1258

6MWD, 6-minute walking distance; TAPSE, tricuspid annular plane systolic excursion; FAC, fraction area change; LVEF, left ventricular ejection fraction; RVSP, right ventricular systolic pressure; LVGLS, left ventricular global longitudinal strain; RVFLS, right ventricular free-wall longitudinal strain; LASr, left atrial reservoir function; LASc, left atrial conduit function; LASp, left atrial pump function; RASr, right atrial reservoir function; RASc, right atrial conduit function; RASp, right atrial pump function.

**Table 4 diagnostics-12-02266-t004:** Results of univariate and multivariate Cox model analysis.

Variables	Univariate Regression			Multivariate Regression		
	Hazard Ratio	95%CI	*p*	Hazard Ratio	95%CI	*p*
Age	1.01	0.975–1.045	0.59			
BSA	62.806	0.348–11,294.669	0.118			
BNP	1.001	1.001–1.002	0.001			
6MWD	0.994	0.991–0.998	0.003	0.995	0.991–0.999	0.02
NYHA	2.062	1.227–3.464	0.006			
LVEF	1.007	0.929–1.092	0.863			
FAC	0.927	0.873–0.985	0.014			
TAPSE	0.9	0.799–1.015	0.087			
TEI	1.819	0.712–4.648	0.212			
RVSP	1.041	1.012–1.070	0.005			
LVGLS	1.295	1.077–1.557	0.006			
RVFLS	1.238	1.080–1.420	0.002			
LASr	0.954	0.898–1.013	0.124			
LASc	0.911	0.833–0.996	0.041			
LASp	1.012	0.891–1.149	0.856			
RASr	0.95	0.904–0.997	0.038			
RASc	0.946	0.869–1.030	0.201			
RASp	0.928	0.861–1.001	0.053			
|LVGLS/RVFLS-1|	4.026	1.872–8.662	<0.001	3.152	1.413–7.034	0.005
|LASr/RASr-1|	1.28	0.800–2.046	0.303			
|LASc/RASc-1|	0.501	0.210–1.197	0.12			
|LASp/RASp-1|	1.441	1.126–1.845	0.004			

6MWD, 6-minute walking distance; TAPSE, tricuspid annular plane systolic excursion; FAC, fraction area change; LVEF, left ventricular ejection fraction; RVSP, right ventricular systolic pressure; LVGLS, left ventricular global longitudinal strain; RVFLS, right ventricular free-wall longitudinal strain; LASr, left atrial reservoir function; LASc, left atrial conduit function; LASp, left atrial pump function; RASr, right atrial reservoir function; RASc, right atrial conduit function; RASp, right atrial pump function.

## Data Availability

The data presented in this study are available on request from the corresponding author. The data are not publicly available due to privacy or ethical restrictions.

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
