# Peer review of "The Balance between the Left and Right Ventricular Deformation Evaluated by Speckle Tracking Echocardiography Is a Great Predictor of the Major Adverse Cardiac Event in Patients with Pulmonary Hypertension"

_diagnostics, 2022, doi:10.3390/diagnostics12092266_

Round 1

Reviewer 1 Report

The paper of Zhang and el. is quite interesting.

The Authors showed that  that |LVGLS/RVFLS-1| is of high clinical and prognostic relevance in pulmonary hypertension patients and indicates the importance of the balance between the left and right ventricular deformation.

The Authors used a proprite statistical methods,The tables and figures are clear.

The referenses are up to date. The  conclusions follow strictly from the results obtained.  My minor remarks is: in my opinion it  will be interesting to introduce  actual and    future  directions of speckle tracking echocardiography.

Author Response

Thank you for your letter and the reviewers’ comments on our manuscript entitled "The balance between the left and right ventricular deformation
evaluated by speckle tracking echocardiography is a great predictor of the
major adverse cardiac event in patients with pulmonary hypertension" (ID:diagnostics-1867686).
Those comments help us both in English and in depth to improve the quality of the paper.

1.We have checked  and modified some non-standard expressions or grammatical errors carefully which we hope meet with approval.

Reviewer 2 Report

It could be interesting to discuss also the consequent dyssynchrony between the two ventricles, due to the retrograde electric stimulus switching from one to another at the apex level or through the septum. This  can be explained anatomically with the absence of a real barrier between the two systems of fibers and , more, by their reciprocal interconnections. 

Author Response

Thank you for your letter and the reviewers’ comments on our manuscript entitled "The balance between the left and right ventricular deformation
evaluated by speckle tracking echocardiography is a great predictor of the
major adverse cardiac event in patients with pulmonary hypertension" (ID:diagnostics-1867686).
Those comments are very helpful for revising and improving our paper, as well as the important guiding significance to other research. The main corrections are in the manuscript and the responds to the reviewers’ comments are as follows.

1.In view of how electrophysiological remodelling affects mechanics, it might be nessessary to include electrophysiological remodelling mechanisms in our  discussion of electrical dyssynchrony and growth in the heart. We have made corrections which we hope meet with approval.

Reviewer 3 Report

The paper is interesting and well written.
The relationship between social isolation and visceral pain is a very relevant topic.
I just suggest to cite in your discussion possible mechanisms underlying the muscle-protective effects (10.1007/s11357-012-9428-4)

Author Response

Thank you for your letter and the reviewers’ comments on our manuscript entitled "The balance between the left and right ventricular deformation
evaluated by speckle tracking echocardiography is a great predictor of the
major adverse cardiac event in patients with pulmonary hypertension" (ID:diagnostics-1867686).
Those comments are very helpful for revising and improving our paper, as well as the important guiding significance to other research. The main corrections are in the manuscript and the responds to the reviewers’ comments are as follows.

1.It might be interesting to include oxidative stress and pro-inflammatory signaling mechanisms in our discussion. We have cited the article you provided which discussed possible mechanisms underlying the muscle-protective effects.
